# Murine Falcor/LL35 lncRNA Contributes to Glucose and Lipid Metabolism In Vitro and In Vivo

**DOI:** 10.3390/biomedicines10061397

**Published:** 2022-06-13

**Authors:** Evgeniya Shcherbinina, Tatiana Abakumova, Daniil Bobrovskiy, Ilia Kurochkin, Ksenia Deinichenko, Elena Stekolshchikova, Nickolay Anikanov, Rustam Ziganshin, Pavel Melnikov, Ekaterina Khrameeva, Maria Logacheva, Timofei Zatsepin, Olga Sergeeva

**Affiliations:** 1Skolkovo Institute of Science and Technology, 121205 Moscow, Russia; evgeniia.shcherbinina@skoltech.ru (E.S.); t.abakumova@skoltech.ru (T.A.); i.kurochkin@skoltech.ru (I.K.); e.stekolschikova@skoltech.ru (E.S.); n.anikanov@skoltech.ru (N.A.); e.khrameeva@skoltech.ru (E.K.); m.logacheva@skoltech.ru (M.L.); t.zatsepin@skoltech.ru (T.Z.); 2Faculty of Bioengineering and Bioinformatics, Lomonosov Moscow State University, 119234 Moscow, Russia; daniilbobrovsky@gmail.com; 3Institute of Biomedical Chemistry, 119121 Moscow, Russia; kseniadey@yandex.ru; 4Shemyakin-Ovchinnikov Institute of Bioorganic Chemistry, 117997 Moscow, Russia; rustam.ziganshin@gmail.com; 5Serbsky National Medical Research Center for Psychiatry and Narcology, 119034 Moscow, Russia; proximopm@gmail.com; 6Department of Chemistry, Lomonosov Moscow State University, 119991 Moscow, Russia

**Keywords:** long non-coding RNA, lipid metabolism, glucose metabolism, hepatocyte, liver

## Abstract

Glucose and lipid metabolism are crucial functional systems in eukaryotes. A large number of experimental studies both in animal models and humans have shown that long non-coding RNAs (lncRNAs) play an important role in glucose and lipid metabolism. Previously, human lncRNA DEANR1/linc00261 was described as a tumor suppressor that regulates a variety of biological processes such as cell proliferation, apoptosis, glucose metabolism and tumorigenesis. Here we report that murine lncRNA Falcor/LL35, a proposed functional analog of human DEANR1/linc00261, is predominantly expressed in murine normal hepatocytes and downregulated in HCC and after partial hepatectomy. The application of high-throughput approaches such as RNA-seq, LC-MS proteomics, lipidomics and metabolomics analysis allowed changes to be found in the transcriptome, proteome, lipidome and metabolome of hepatocytes after LL35 depletion. We revealed that LL35 is involved in the regulation of glycolysis and lipid biosynthesis in vitro and in vivo. Moreover, LL35 affects Notch and NF-κB signaling pathways in normal hepatocytes. All observed changes result in the decrease in the proliferation and migration of hepatocytes. We demonstrated similar phenotype changes between murine LL35 and human linc00261 depletion in vitro and in vivo that opens the opportunity to translate results for LL35 from a liver murine model to possible functions of human lncRNA linc00261.

## 1. Introduction

Although most of the mammalian genome is transcribed into RNA, only a small fraction of these RNA encode proteins [1]. In the last decade non-coding RNAs (ncRNAs) have received considerable attention as possible prognostic makers and regulators of progression for many diseases [2]. Thus, most studies have investigated ncRNAs in cancer cells and tissues, while much less is known about the role of ncRNAs in physiological processes—there are still many undiscovered roles of ncRNA. Long non-coding RNAs (lncRNA) are defined as ncRNA molecules longer than 200 nt; their expression is usually tissue- or stimuli-specific [3]. LncRNA play an essential role in the regulation of various cellular processes, including replication, transcription, translation, chromatin remodeling and post-translational modification of proteins [4]. In addition, a number of lncRNA are involved in the regulation of metabolic processes such as glucose and lipid metabolism, oxidative stress and mitochondrial function [5]. As well as humans, lncRNA are also found in other mammals [6,7], insects [8], nematodes [9] and fish [10]. Cross-species comparisons have shown that lncRNA may contain short conserved regions [11], but in most cases they are better defined as functional analogs due to poor sequence similarity. The search for functional analogs in different species is an important task since the use of animal models allows studying lncRNA in vivo [12]. 

Human lncRNA DEANR1/linc00261 is involved in many cellular processes, including ones crucial for cancer development [13,14,15,16]. Linc00261 expression is downregulated in several cancer types, such as HCC, pancreatic, gastric, colorectal, lung and breast cancers [17]. Linc00261 as a tumor suppressor inhibits the growth of cancer cells primarily by inhibiting cell proliferation and stimulating cell apoptosis [18], and also reduces the invasiveness of cancer cells by inhibiting the epithelial–mesenchymal transition (EMT) [19,20]. Expression of linc00261 also enhances the antitumor effect of cisplatin in colon cancer by reducing the level of β-catenin in the nucleus [21]. Linc00261 can inhibit the activation of the Wnt signaling pathway, thereby promoting degradation of β-catenin and blocking the translocation of β-catenin from the cytoplasm to the nucleus [21]. Moreover, linc00261 can act as ceRNAs, which sponge miRNA and affect the mRNA levels of target genes. Many target genes of linc00261 participate in multiple signaling pathways, such as the Wnt/β-catenin signaling pathway, p38 MAPK signaling pathway and Notch signaling pathway [17]. Murine lncRNA Falcor/LL35 (NCBI—9030622O22Rik) has been defined as a possible functional analog of human lncRNA DEANR1/linc00261, which is predominantly expressed in murine lungs and liver, intestine and pancreas [22,23]. Using knockout mice, Swarr et al. described the functional significance of LL35–Foxa2 (transcription factor) regulatory loop during regeneration after lung injury. LL35 deletion in mice disrupted the integrity of the airway’s epithelial barrier and peribronchial inflammation, and impaired epithelial regeneration after injury in lungs [23]. Regarding the role of LL35 in the liver, we have demonstrated that LL35 expression is downregulated by 70% in mouse liver two weeks after the induction of fibrosis, followed by a partial recovery of LL35 expression after several weeks [22]. At the same time, the function of LL35 in the liver and the effect of its suppression on the functioning of hepatocytes remain unknown.

In this work we demonstrated the prevalent expression of lncRNA LL35 in normal hepatocytes in comparison to cancer and proliferating cells in vitro and in vivo and studied the roles of LL35 in hepatocytes. We found that the depletion of LL35 in vitro and in vivo caused changes in the mRNA levels of genes involved in metabolism, glutathione conjugation, biological oxidations and sulfur amino acid metabolism. Lipidome analysis demonstrated the decrease in plasmanyl-/plasmenylphosphatidylcholines (O-PC/P-PC) both in the cell line and liver, while phosphatidylethanolamines (PE) and hexosylceramides (HexCer) increased only in cells, and diradylglycerols (DG), ceramides (Cer;O2) and lysophosphatidylcholines (LPC) increased only in the liver. We found involvement of lncRNA LL35 in glycolysis in vitro and in vivo and demonstrated a reduction in the extracellular acidification rate after LL35 depletion in vitro. We demonstrated more pronounced phenotype similarity between murine LL35 and human linc00261 depletion in vitro and in vivo. This result clearly opens the opportunity to translate results from a liver-specific KD murine model to novel roles of human lncRNA linc00261 in normal physiology. 

## 2. Materials and Methods

### 2.1. Cell Culture and Transfection

Experiments were performed using normal hepatocyte cell line AML12 (ATCC^®^CRL-2254, Manassas, VA, USA), hepatoma cell lines Hepa 1-6 (ATCC^®^CRL-1830, VA, USA) and Hepa-1c1c7 (ATCC^®^CRL-2026, VA, USA). Cells were cultured in DMEM/F12 (Gibco, Waltham, MA, USA), 10% FBS (Gibco, Waltham, MA, USA), 2 mM L-glutamine, 1% penicillin and 1% streptomycin (10,000 U/mL, Gibco, MA, USA) at 37 °C and under 5% CO_2_. Cells were split when they reached 80% confluency. Cells were transfected with antisense oligonucleotides (ASOs) using Lipofectamine RNAiMAX (Invitrogen, Waltham, MA, USA) according to the manufacturer’s protocol. Cells were analyzed 48 h after transfection. For insulin stimulation experiment AML12 cells were cultured for 2 h in reduced serum Opti-MEM media (Gibco, MA, USA) and then treated with 100 nM insulin solution (Paneco, Moscow, Russia).

### 2.2. Antisense Oligonucleotides (ASOs) Description

Design, validation and selection of the most efficient ASOs targeting LL35 (NCBI—9030622O22Rik) for cells’ treatment are described previously in [22]. LL35 downregulation was performed using a mix of 5 ASOs (Appendix A) at the final concentration of 20 nM for each ASO. ASO specific to Firefly Luciferase gene was used as a control at 100 nM concentration. For targeted delivery to the mouse liver, we synthesized inhouse 3′-GalNAc conjugated ASO [24] with the same core sequences as described above (Appendix A).

### 2.3. Animal Care and Treatments

Balb/c mice for this study were purchased from “Stolbovaya” Scientific Center of Biomedical Technologies of the Federal Medical and Biological Agency. Study protocols were approved by the Bioethics Committee of the Institute of Developmental Biology, all animals received human care according to the National Institute of Health guidelines. Mice were maintained at 22 °C using a 12 h light to 12 h dark cycle. Hepatocellular carcinoma was induced using plasmids encoding human ΔN90-β-catenin, human MET and Sleeping Beauty transposase as described in [25]. Partial hepatectomy was performed as described in [26]. 

For the depletion of LL35 in the liver, a mix of ASO was diluted in sterile saline and injected intravenously via the tail vein at the following total ASO doses: 25 mg/kg, 50 mg/kg, 100 mg/kg 150 mg/kg and 200 mg/kg (three mice per group). Mice were sacrificed at day 2 and 5 after ASOs injection and serum and liver samples were collected for analysis. Serum was collected by cardiac puncture, followed by centrifugation at 1700× *g* for 20 min. Serum biochemical analysis was performed in Paster Laboratories (Moscow, Russia). 

### 2.4. Insulin Tolerance Test

Mice were fasted overnight. Soluble human insulin (Paneco, Moscow, Russia) at 1 U/kg was administered by intraperitoneal injection (three mice per group). Glucose was monitored by tail bleeding at 0, 15, 30, 45 and 60 min using a glucometer. Mice were sacrificed after the last glucose measurement.

### 2.5. Histological Analysis

Freshly collected liver samples were dehydrated and embedded into paraffin in the Tissue TEK VIP 5 Jr (Sakura, Tokyo, Japan). Eight-micrometer-thick sections were subjected to hematoxylin and eosin (Abcam, Cambridge, UK) and Oil Red O (Sigma-Aldrich, St. Louis, MO, USA) staining according to the manufacturer’s instructions.

### 2.6. RNA Isolation, cDNA Synthesis and RT-qPCR

Total RNA was isolated from cells or liver tissue using TRIzol (Thermo Fisher Scientific, Waltham, MA, USA), followed by precipitation with isopropanol, according to the manufacturer’s instructions. Liver tissue samples were homogenized in TRIzol using the Precellys^®^ homogenizer. A total of 0.5–1 μg of total RNA was further treated with DNase I (Thermo Fisher Scientific, Waltham, MA, USA) and supplied with RiboLock RNase Inhibitor (40 U/μL) to the final concentration 0.4 U/μL. cDNA was generated using a Maxima First Strand cDNA Synthesis Kit (Thermo Fisher Scientific, Waltham, MA, USA) according to manufacturer’s protocol. RNA levels were assessed by qPCR using qPCRmix-HS LowROX Kit (Evrogen, Moscow, Russia) in the Cycler CFX96 Touch Real-Time PCR detection system (Bio-Rad Laboratories, Hercules, CA, USA). The RNA levels of interest were normalized to the level of the mouse housekeeping genes GAPDH (glyceraldehyde 3-phosphate dehydrogenase) or ACTB (β-actin) and to the average value of the control group where needed. RT-qPCR was performed using specific primers listed in Appendix A.

### 2.7. Western Blotting

Cell protein extracts were prepared from LL35 KD and control cell samples using RIPA Lysis and Extraction Buffer (Thermo Fisher Scientific, Waltham, MA, USA) and supplied with 1× Halt™ Protease Inhibitor Cocktail (Thermo Fisher Scientific, Waltham, MA, USA) according to manufacturer’s protocol. Liver samples were homogenized using the Precellys^®^ homogenizer and protein extracts were prepared using RIPA Lysis and Extraction Buffer (Thermo Fisher Scientific, Waltham, MA, USA) supplied with 1× Halt™ Protease Inhibitor Cocktail (Thermo Fisher Scientific, Waltham, MA, USA), 0.05% Triton X-100 (Helicon, Moscow, Russia), 1 mM dithiothreitol (DTT) (Helicon, Moscow, Russia), 0.2 mM phenylmethylsulfonylfluoride (PMSF) (Sigma-Aldrich, St. Louis, MO, USA). The total protein concentration in the lysate was measured using Pierce™ BCA protein assay kit (Thermo Fisher Scientific, Waltham, MA, USA) or Pierce™ Coomassie Plus (Bradford) Assay Kit (Thermo Fisher Scientific, Waltham, MA, USA). An amount of 20–60 µg of total protein samples were run on 10% SDS–polyacrylamide gels and transferred to PVDF membranes (Bio-Rad Laboratories Inc., CA, USA) using Mini Trans-Blot^®^Cell and Criterion™ Blotter (Bio-Rad Laboratories Inc., CA, USA) (standard protocol) and incubated with primary antibodies (listed in Appendix A) overnight at 4 °C or for 1 h at room temperature after membrane blocking in TBS/Tween with 5% bovine serum albumin (Sigma-Aldrich, MO, USA) at 4 °C overnight. For protein bands’ visualization we used the appropriate HRP-linked secondary antibodies and Clarity™ Western ECL Blotting Substrates (Bio-Rad Laboratories Inc., CA, USA). Western blot images were quantified with ImageJ software according to its standard protocol [27].

### 2.8. Cell Viability Assay

AML12 cells were plated into 48-well plates in four replicates, ~22 × 10^3^ cells per well, and transfected with LL35 ASOs mix or control Luc ASO (final concentration 20 nM of each ASO) using Lipofectamine RNAimax (Invitrogen, Waltham, MA, USA) according to the manufacturer’s protocol. The viability of the cells was measured at 24, 48 and 72 h after initial transfection using CellTiter 96^®^Aqueous One Solution Cell Proliferation Assay (MTS) (Promega, Madison, WI, USA), followed by 3 h incubation at 37 °C according to manufacturer’s protocol. Then, the fluorescent signal was measured using a Varioscan Microplate reader with a 490 nm filter (Thermo Fisher Scientific, Waltham, MA, USA). The obtained data were presented on a plot of optical density (O.D.) versus number of days, showing mean ± SD. Assessment of the significance of the difference in viability between cells with LL35 knockdown and control cells was performed using the multiple *t*-test. 

### 2.9. Wound Healing and Migration Assay

AML12 cells transfected with LL35 ASOs or control Luc ASO using Lipofectamine RNAimax (Invitrogen, Waltham, MA, USA) were cultured in 6-well plates until confluence reached 70–80%, then several wounds with a width of approximately 1.5 mm were introduced in cell monolayers using a pipette tip. To trace the wound closure, pictures were taken at 48 h and 72 h after wound introduction in 6–8 replicas per each sample. Total wounding area was measured using ImageJ software [27]. 

### 2.10. Cell Cycle Analysis with Flow Cytometry

AML12 cells were plated into 6-well plates in triplicate, ~3 × 10^5^ cells per well, and transfected with LL35 ASO mix or control Luc ASO (final concentration 20 nM of each ASO) using Lipofectamine RNAimax (Invitrogen, Waltham, MA, USA). On the second day after transfection, cells were collected, washed twice with 1× PBS and fixed overnight at 4 °C with 2 mL of 70% ethanol. After fixation, cells were washed with 1× PBS and then re-suspended in 0.5 mL of 1 × PBS with 5 μg/mL RNase A and 30 μg/mL PI (propidium iodide). Cell cycle measurement was performed by Flow Cytometer Bio-Rad ZE5 (Bio-Rad Laboratories Inc., CA, USA). The flow cytometry results were analyzed using FlowJo™ v10.8 Software (BD Life Sciences, Franklin Lakes, NJ, USA).

### 2.11. RNA-Seq Data Processing and Analysis

For AML12 cells’ transcriptome analysis, we used ~1,2 × 10^6^ cells per sample after 48 h of ASO-mediated LL35 knockdown or control Luc knockdown; 4 replicates per experiment were used. For liver transcriptome analysis we collected samples 5 days after LL35 ASOs or Luc ASO injection; 3 mice per experiment were used. Total RNA from cells and liver samples was isolated using a TRIzol (Thermo Fisher Scientific, Waltham, MA, USA), followed by precipitation with isopropanol, according to the manufacturer’s protocol. Four micrograms of total RNA were used for further sequencing library preparation using NEBNext Ultra II Directional RNA Library Prep Kit for Illumina (NEB 7760, New England Biolabs, Ipswich, MA, USA), with previous fragmentation and rRNA depletion. Detailed protocol for library preparation is described in Appendix A. Libraries were sequenced using a HiSeq4000 (Illumina, San Diego, CA, USA) instrument in 5 nt single-read mode. bcl2fastq2 software (Illumina, San Diego, USA) was used for conversion to fastq format. 

RNA-seq reads were aligned to mouse genome using STAR v2.5.3a with default settings, except–quantMode GeneCounts, genome annotations were obtained from Ensembl. We used R package DESeq2 for gene counts processing and RLE method for further normalization. Differential expression analysis was performed using DESeq2 package based on the Wald test. Genes were considered as differentially expressed if they passed the following thresholds: FDR < 0.1 and |log2foldChange| > 0.8. For differentially expressed genes we performed functional enrichment analysis with PANTHER [28] using BP/MF/CC ontology and Reactome pathways as annotation set.

### 2.12. LC-MS-Based Proteomics

Cells were cultivated in 6-well plate and collected at the second day after LL35 knockdown, washed twice with 1× PBS and processed for further LC/MS analysis. Reduction, alkylation and digestion of the proteins in solution were performed as described previously [29,30] with minor modifications. Before analysis, peptides were dissolved in 2% acetonitrile/0.1% TFA solution and sonicated for 2 min in ultrasonic water bath. Detailed sample processing protocol to obtain proteomics data is described in Appendix A.

We used MaxQuant [31] and Perseus [32] for raw spectra processing, search was conducted against *Mus musculus* Uniprot Tremble database (version from 03.2018). For MaxQuant search the default parameters included Trypsin/p protease specificity with maximum 2 missed cleavages. Met oxidation, protein N-term acetylation and Asn/Gln deamidation were used as variable modifications, while Carbamidomethyl Cys were used as a fixed modification with maximum 5 modifications per peptide; PSM and protein FDR were applied as 1%. All runs were processed in Perseus. Proteins with maxLFQ values in at least 3 out of 6 LC-MS runs were chosen for further analysis. Missing values for selected proteins were imputed from normal distribution with 0.3 intensity distribution sigma width and 1.8 intensity distribution center downshift. Statistically significantly changed proteins after LL35 knockdown were selected based on two-sample t-test with permutation-based FDR 5%. 

### 2.13. Seahorse Extracellular Flux Analysis

The Seahorse XF96 Extracellular Flux Analyzer (Agilent, Santa Clara, CA, USA) was used to measure the respiration activity of AML12 cells. Cells were seeded at the density of ~12 × 10^3^ cells per well in a XF96 plate, the next day cells were transfected with LL35 or Luc ASOs. The Glycolysis Stress Test and Mito Stress Test were performed at the second day after transfection according to manufacturer’s protocol. All Seahorse XF96 measurements were normalized to the protein contents in each well, measured by Pierce™ BCA protein assay kit (Thermo Fisher Scientific, Waltham, MA, USA). The relative levels of glycolysis, glycolytic capacity and glycolytic reserve were calculated based on ECAR (extracellular acidification rate) data obtained in the Glycolysis Stress Test using Seahorse Wave software for XF analyzers. Non-mitochondrial consumption, basal respiration, maximum respiration, proton leak, ATP production and spare respiratory capacity were calculated based on OCR (oxygen consumption rate) data obtained during Mito Stress Test. 

### 2.14. Mitochondria Staining with TMRE and Mito Green FM

For mitochondria staining, AML12 cells were cultivated on poly-L-lysine-coated microscopy glasses. Both TMRE (tetramethylrhodamine, ethyl ester) (Thermo Fisher Scientific, Waltham, MA, USA) and MitoTracker Green FM (Thermo Fisher Scientific, Waltham, MA, USA) were used at day 2 after LL35 knockdown according to manufacturer’s protocol. Imaging was performed by Nikon A1+MP confocal imaging system (Nikon, Minato City, Japan). 

### 2.15. Lipid Staining with Oil Red O for AML12 

AML12 cells were cultivated on 35-mm confocal dishes (VWR International, Radnor, PA, USA) and Oil Red O (Sigma-Aldrich, St. Louis, MO, USA) staining was performed 48 h after ASO-mediated LL35 knockdown or control Luc knockdown. Briefly, Oil Red O powder was dissolved in 100% isopropanol to obtain a 30% solution. Then 3 parts of 30% Oil Red O solution were added to 2 parts of water, mixed and filtered to obtain a working solution. Cells were washed twice with 1× PBS and fixed for 30 min in 4% formaldehyde. After two washes with water and 5 min incubation in 60% isopropanol, cells were stained with Oil Red O working solution for 20 min followed by staining of cell nuclei with DAPI (Invitrogen, Waltham, MA, USA). After washing off the DAPI, cell imaging was performed by Nikon A1+MP confocal imaging system (Nikon, Japan). 

### 2.16. Cell Lines and Liver Tissue Extraction Protocol and Liquid Chromatography/Mass Spectrometry Analysis for Lipids and Metabolites Analysis 

AML12 cells were plated into 150-mm cell culture dish (1 dish per biological replica, 3 replicas per condition). At day 2 after transfection with ASOs mix (20 nM each) for LL35 or Luc using Lipofectamine RNAiMax (Invitrogen, Waltham, MA, USA), cells were collected and washed with PBS solution twice, supernatant was discarded and cell pellets were frozen at −80 °C until extraction. On extraction day, cells were thawed on ice for several minutes and randomized. For liver samples (3 samples per group), 10–15 mg liver tissue pieces were dissected from the frozen tissue samples on dry ice, weighed, randomized and transferred to cooled 2 mL Precellys tubes with 6 zirconium oxide beads (Bertin Technologies, Montigny-le-Bretonneux, France). Extraction blank samples were inserted in the end of main batch, consisting of an empty tube without cell or tissue sample. The extraction buffer recipe and further samples’ preparation for LC-MS analysis are provided in Appendix A. Briefly, phase separation was induced by adding 700 μL of water/methanol (3/1, v/v) mixture with the following standards (6.7 µg/mL): L-Glutamic acid-^13^C5 (604860, Sigma-Aldrich, MO, USA), methionine-methyl-13C, d3 (299154, Sigma-Aldrich, MO, USA), GMP-^15^N5 (662674, Sigma-Aldrich, MO, USA). A total of 540 μL (for cells) or 300 μL (for liver) of the upper layer, containing most of the lipids, was collected to an Eppendorf tube. In total, 900 μL (for cells) or 1000 μL (for liver) of polar lower phase were collected in a separate Eppendorf tube. Solvents were evaporated under reduced pressure at 30 °C in the Speed Vac concentrator (mode V-HV, Eppendorf Concentrator plus Complete System, Hamburg, Germany). Dried samples were stored at −80 °C until mass spectrometry analysis. 

Liquid chromatography/mass spectrometry system consisted of Waters Acquity I-Class UPLC system (Waters, Manchester, UK) and Q Exactive orbitrap mass spectrometer (Thermo Fisher Scientific, Waltham, MA, USA) equipped with a heated electro-spray ionization (HESI) probe used for lipids’ and metabolites’ data acquisition from AML12 cells and liver tissue samples. We conducted a lipid separation at 60 °C on a reverse phase ACQUITY UPLC BEH C8 Column (2.1 × 100 mm, 1.7 μm, Waters co., Milford, MA, USA) equipped with Vanguard pre-column at flow rate of 0.4 mL/min, while polar metabolites’ separation was performed at 40 °C on a ZIC-HILIC Column (2.1 × 100 mm, 3.5 μm, SeQuant, Merck, Germany) equipped with pre-column at a flow rate of 0.4 mL/min. Further details related to lipids’ and metabolites’ separation as well as mass spectrometry parameters are described in Appendix A.

### 2.17. Post-Acquisition Processing and Statistical Analysis of Lipidome and Metabolome Data

Vendor-format files (.raw) were imported to Progenesis software (version 2.3, Non-linear Dynamics, Newcastle, UK), then features were automatically extracted, aligned and deconvoluted with default software settings and following adducts: M+H, M+NH_4_, M+Na, M+K, M+H-H_2_O, M+H-2H_2_O, M+2Na, M+H+Na, M+2H+Na for positive polarity and M-H, M+FA-H, M+AA-H, M-H_2_O-H for the negative one. Alignment quality was more than 92% for all biological and technical samples and more than 85% for extraction blanks. Data (rt-m/z matrix) was exported to .csv format, then subsequent normalizations and data analysis were carried out in R environment (v.4.1.2). After several filtration steps based on missing values, retention time and coefficient of variation, lipid features from all the samples were normalized by the median abundances of internal standards; features from liver samples were also normalized by sample weight. The resulting abundances were log2-transformed. Features with mean abundance in samples < mean abundance in blank samples + 2 (in log2-scale) were removed from the analysis.

Putative annotation of lipids’ and metabolites’ species was performed against LipidMaps (https://www.lipidmaps.org/, accessed on 31 March 2022) and Human Metabolome Database (https://hmdb.ca/, accessed on 15 March 2022) databases. Significantly enriched lipids were used to create an inclusion list for DDA-mode analysis. LipoStar [33] software was employed for confirmation of lipid classes. To visualize variation in lipid and metabolite content across samples, we performed principal component analysis (PCA). To assess the effect of the LL35 knockdown on the abundance of different lipid classes, we applied fast gene set enrichment analysis (fgsea) [34] with lipid classes in place of gene sets. To rank lipid features for this procedure, we used log2 fold changes (logFC). We subsequently performed confirmation of features from lipid classes for which significant alteration (fgsea FDR < 5%) was observed. Only lipid features with MS2-confirmed annotations were used for further fgsea recalculation. Only lipid classes with FDR< 10% both before and after confirmation were considered to be significantly influenced by LL35 knockdown. The alterations in the abundance of metabolites were also determined using fgsea. In this case it was performed on pathways from KEGG [35] and on a subset of 99 SMPDB [36] metabolite sets used by MetaboAnalyst [37], ranking features by their logFC. To identify the metabolites responsible for differences between samples, we also used sparse PCA [38] as implemented in R package mixOmics [39] and pairwise Welch t-test with unequal variance assumption and Benjamini–Hochberg (BH) correction for multiple testing.

### 2.18. Statistical Analysis

All diagrams presented here are based on at least three independent experiments. Statistical processing of obtained data was performed using the GraphPad Prism software (version 8.0.1) (GraphPad Holdings, LLC, San Diego, CA, USA) with multiple t-tests. The data were considered statistically significant at *p*-value < 0.05. Venn diagram was built using Venny 2.1 software [40].

## 3. Results

### 3.1. LL35 Expression in Murine Hepatocytes and Liver

Our previous data demonstrated the prevalent expression of LL35 lncRNA gene (9020622O22Rik) in murine liver and lungs [22]. To estimate the role of LL35 in the physiology of hepatocytes we compared the expression levels of LL35 in a normal murine hepatocyte cell line (AML12) and two hepatoma cell lines (Hepa 1-6 and Hepa-1c1c7) by RT-qPCR. We revealed that the LL35 RNA level is much higher in AML12 in comparison with both hepatoma cell lines (Figure 1A). A lot of studies have shown that the deregulation of lncRNAs is associated with the development and progression of various cancers, which makes them suitable biomarkers for cancer diagnosis and prognosis [41]. 

The downregulation of LL35 in cancer cells pointed to checked LL35 RNA levels in the liver under proliferative conditions—a model of hepatocellular carcinoma and after partial hepatectomy. First, we evaluated LL35 expression in a hepatocellular carcinoma (HCC) mouse model induced by plasmids encoding human ΔN90-β-catenin, human MET and Sleeping Beauty transposase [25]. This model is the closest one to human HCC [25]. We observed that under these conditions LL35 expression is decreased in comparison to a normal liver (Figure 1B). 

As a lot of lncRNA change expression during liver regeneration [42], we measured LL35 RNA levels after partial hepatectomy [26] and found its downregulation 2 h after surgery followed by an increase in the RNA level during the liver’s regeneration (Figure 1C). In addition, in our previous study we demonstrated ~70% decrease in the LL35 lncRNA level in mouse liver two weeks after the induction of fibrosis with carbon tetrachloride [22]. Taking together the data obtained using cell lines and mouse models, we considered that LL35 can be functionally significant in a normal murine liver and may be a probable biomarker in liver diseases. 

### 3.2. LL35 Downregulation In Vitro and In Vivo

To study the role of LL35 we used antisense oligonucleotides (ASOs) to knockdown LL35 in vitro and in vivo. In previous work [22] we tested and selected a mix of five ASOs (20 nM each) for the specific downregulation of LL35 in AML12 cells. On day 2 after transfection, the decrease in LL35 RNA expression exceeded 80% (Appendix A). For LL35 downregulation in vivo we synthesized five GalNAc conjugates with ASOs for the targeted delivery in the murine liver (Appendix A, Appendix A). Triple GalNac residues conjugated to ASO or siRNA drive the efficient delivery to the liver after IV or SC injection [24]. GalNAc-ASO conjugates were injected in the murine tail vein at different concentrations: 25 mg/kg, 50 mg/kg, 100 mg/kg, 150 mg/kg and 200 mg/kg in total and the efficacy of LL35 RNA inhibition was analyzed at days 2 and 5 by RT-qPCR (Appendix A). On day 2 after ASO’s injection, the knockdown efficacy was ~63%. Meanwhile, on day 5. LL35 RNA expression in mice that received doses 150 mg/kg and 200 mg/kg decreased to 19% and 16%, respectively, in comparison with mice injected with control Luc GalNAc-ASO. Thus, we chose day 5 after the injection with the 150 mg/kg ASO mix for further experiments. (Figure 1D). To explore the effects of LL35 lncRNA downregulation after the administration of GalNAc-ASO conjugates in the murine liver, we performed a morphological study using hematoxylin–eosin (H&E) staining of the liver samples. According to our results, no pathological changes were observed in the LL35 ASO mice in comparison with the Luc control group of mice (Appendix A). Analysis of the animal blood demonstrated that levels of ALT, AST, ALP and a majority of the other key factors were not changed (Appendix A). Moreover, no significant weight changes or differences in the behavior of the mice were observed between the groups, indicating that the administration of GalNAc-ASO was well tolerated.

### 3.3. Analysis of Gene and Protein Expression after LL35 Knockdown In Vitro and In Vivo 

RNA sequencing was used to investigate the changes in mRNA profiles in AML12 cells on day 2 and in liver tissue on day 5 after LL35 knockdown (Appendix A, Appendix A). We calculated differential expression (|log2foldchange| > 0.8, adjusted *p*-value < 0.1) and found that 796 genes significantly changed their expression in vitro and 170 genes in vivo in comparison with the control Luc ASOs, while only five genes were common between the two sets (Figure 2A). Among 796 differentially expressed genes in AML12 cells, 523 were upregulated and 273 were downregulated; meanwhile, in the liver tissue, 63 genes showed significant upregulation and 107 downregulation. 

We conducted a PANTHER Reactome pathway analysis [28] for differentially expressed genes after LL35 downregulation in the AML12 cells and mouse liver. In the AML12 cells, the most enriched pathways (*p*-value < 0.05) are related to the cell cycle, rRNA processing and lipid metabolism (glycerophospholipid biosynthesis, complex I biogenesis, metabolism of steroids) (Figure 2B). In the liver tissue after LL35 knockdown, no changes were observed in the pathways related to the cell cycle, which can be a result of the slow turnover of hepatocytes [43]. At the same time, we found changes in several pathways (*p*-value < 0.05) such as metabolism, glutathione conjugation, biological oxidations and sulfur amino acid metabolism, which correlated with the changes in vitro (Figure 2C). 

Then we performed a gene ontology analysis by three gene attributes: molecular function (MF) (Appendix A), cellular component (CC) (Appendix A) and biological process (BP) (Appendix A). Enriched data showed similar effects of LL35 knockdown in vitro and in vivo in the reaction to the endogenous stimulus and signaling receptor activity. An interesting difference was found in the analysis of the cellular components’ changes: for liver tissue we saw an abundance of changes in nuclear compartments, while in AML12 cells there were changes in the compartments associated with intracellular contacts. 

To evaluate the obtained PANTHER Reactome pathways and confirm the transcriptome data, we performed RT-qPCR analysis of the genes involved in cell cycle regulation, lipid biosynthesis and metabolism after LL35 depletion in vitro and in vivo (Figure 3A). The observed changes correlate with the data of transcriptome analysis and confirm the results of the PANTHER analysis.

Using a liquid chromatography/mass spectrometry, we performed a proteome analysis of AML12 cells at day 2 after LL35 depletion. In total, 232 proteins significantly changed their expression level (cutoff 2) after LL35 knockdown in comparison with the control; among them, 130 were downregulated and 102 upregulated. Enrichment analysis by Reactome pathways using PANTHER revealed changes in signaling transduction, protein modification, translation and phospholipid metabolism, which correlates with transcriptome data for AML12 cells (Figure 3B).

### 3.4. LL35 Downregulation Causes Changes in Lipidome and Metabolome Both in AML12 Cells and in Murine Liver

Based on the data of transcriptome and proteome analysis concerning the changes in metabolism and lipid biosynthesis, we performed a high-throughput lipidome and metabolome analysis in vitro and in vivo after LL35 knockdown using liquid chromatography/mass spectrometry. To visualize the variation in lipid and metabolite content across samples, we performed a principal component analysis (PCA) (Figure 4A). Mass spectrometry analyses generated abundances for 294 polar metabolite and 1111 lipid features from the cell culture and 382 polar metabolite and 661 lipid features from the liver tissue putatively annotated (Appendix A). Visualization of the liver samples after LL35 KD demonstrated more variations in comparison to the cell samples (Figure 4A), which can be a result of the several cell types in the liver. 

Group-based analysis assessing the significance of the treatment effect at the level of individual lipid classes revealed a number of classes affected by LL35 depletion in vitro and in vivo. All features from these lipid classes were then subject to additional MS2 confirmation and only lipid classes with fgsea FDR < 10% both before and after confirmation were considered to be significantly dependent on LL35 knockdown. This approach revealed that the abundance of plasmanyl-/plasmenylphosphatidylcholines (O-PC/P-PC) decreased in both AML12 and the murine liver (Figure 4B,C). Furthermore, the abundances of phosphatidylethanolamines (PE) and hexosylceramides (HexCer) were increased only in cell culture, while diradylglycerols (DG), ceramides (Cer;O2) and lysophosphatidylcholines (LPC) showed significant increases in the liver (Figure 4B,C). We checked the mRNA level of enzymes involved in lipid metabolism after LL35 depletion in vitro and in vivo and found the dysregulation of expression for several genes involved in DG, PE and PC biosynthesis (Figure 4D). 

Metabolome analysis revealed changes in purine and glutathione metabolism and the biosynthesis of amino acids after LL35 depletion in vitro, which was supported by a sparse PCA (Figure 4E). Using this method, we selected 30 features for the first principal component; out of these 30 features, 10 turned out to be involved in purine metabolism (MetaboAnalyst Over Representation Analysis, FDR = 0.024). We could not apply a sparse PCA to the liver samples’ analysis since the knockdown and control groups were harder to distinguish even by normal PCA. The obtained data for the metabolome in vitro correlates with the transcriptome and proteome analysis of LL35 depleted cells. 

For the additional quantitative and qualitative measurements of neutral triglycerides and lipids and lipid droplet formation [44] we performed an oil Red O staining. While we did not observe significant difference in lipid droplets between liver samples with LL35 knockdown and the control liver, AML12 cells with LL35 depletion showed an increase in the accumulation of lipids in comparison with the control cells (Appendix A). The obtained micrograph for in vitro data correlates with lipidome analysis. In vivo studies probably need A longer KD to observe phenotype changes. On the other hand, A prolonged KD in vivo will lead to multiple secondary changes.

### 3.5. Inhibition of LL35 lncRNA Interferes with Glucose Metabolism In Vitro and In Vivo

Recently it was demonstrated that the LL35 functional analog in humans, linc00261, participates in the regulation of pancreatic cancer glycolysis and proliferation and also induces cell cycle arrest and apoptosis [13]. Glucose metabolic pathways are associated with lipid metabolism, notably through de novo synthesis of fatty acids and glycerol [45]. Based on these data we decided to analyze glucose metabolism in vitro and in vivo after LL35 depletion. 

Total cellular ATP is predominantly generated by glycolysis and mitochondrial oxidative phosphorylation. To assess the possible involvement of LL35 in ATP synthesis we measured oxygen consumption rate (OCR) (an indicator of mitochondrial oxidative phosphorylation) and extracellular acidification rate (ECAR) (an indicator of glycolysis) levels at day 2 after LL35 knockdown in AML12 cells using a Seahorse analyzer. We did not observe significant changes in OCR levels after LL35 inhibition (Appendix A), which indicates a normal electron flux through the mitochondrial electron transport chain. For additional characterization of the mitochondrial status, we performed TMRE and Mito Green FM staining, which showed no difference in the number of active and total mitochondria between LL35 knockdown and control cells (Appendix A). Taken together these data confirm that LL35 lncRNA is not involved in the normal functioning of mitochondria in vitro. While OCR levels remain unchanged during LL35 inhibition in AML12 cells, ECAR levels are significantly reduced, which means a decrease in glycolytic flux (Figure 5A). In particular, LL35 downregulation results in a decreased glycolytic capacity, glycolytic reserve and non-glycolytic acidification (Figure 5B). To study this effect, we measured RNA levels by RT-qPCR for multiple genes involved in the main pathways of glucose metabolism, such as glycolysis, gluconeogenesis, pentose phosphate pathway, glutaminolysis, glycogen synthesis and acetyl-CoA synthesis pathways after LL35 knockdown. We observed changes in the expression of Glut1 (upregulated) and Glut2 (downregulated) glucose transporters, downregulation of Pdk1 and upregulation of several other genes involved in glycolysis: Eno2, Pfkfb2, Pfkfb3, Hk2 and Ldha (Figure 5C). Expression levels of the main gluconeogenesis genes G6pc and Pepck were also significantly increased after LL35 depletion in vitro. Moreover, Pepck protein upregulation was confirmed by western blot (Figure 5D). We also observed an increase in the expression of four important transcription factors Myc, Pgc1α, Egr1 and Sirt1, regulating genes of glucose and lipids’ metabolism, while FoxJ1’s expression level was downregulated after LL35 inhibition in AML12 cells (Figure 5C). All these data confirmed the involvement of LL35 lncRNA in the regulation of the glucose metabolism, but possible molecular mechanisms are still under investigation.

To verify the involvement of LL35 lncRNA in the glucose metabolism in vivo, we performed an insulin tolerance test. Insulin is the master regulator of glucose, lipid and protein metabolism, and, together with glucagon, maintains normal glucose concentration [46]. We monitored the glucose blood level in mice at day 5 after LL35 knockdown each 15 min for 1 h after insulin injection (1 U/kg) and found that the glucose blood level in mice with LL35 KD was 1.5–2 times higher than in control mice (Figure 6A). Elevated glucose suggests that LL35 depletion results in a decreased insulin response. Analysis of the animal blood after the insulin injection demonstrated the compensation of the effects found for the LL35 depletion. LL35 depletion in vivo resulted in a decrease in direct bilirubin and elevated levels of cholesterol, inorganic phosphorus and AST/ALT ratio (Figure 6B). The injection of insulin caused the absence of the differences between the LL35 ASO mice and control mice. Since insulin is a master regulator that maintains glucose metabolism in mammals, we have checked whether LL35 knockdown causes insulin resistance in vitro. Insulin leads to the activation of the phosphatidylinositol 3-kinase (PI3K)/Akt pathway by autophosphorylation of the β subunit of IR or IGF1 [47,48]. We measured the pAKT1/AKT1 proteins’ ratio before and 1 h after cell stimulation with 100 nM insulin. The level of AKT1 phosphorylation dramatically decreased (pAKT1/AKT1 before insulin stimuli was 3.5 times lower) after LL35 depletion in comparison to the control cells, but insulin addition restored AKT1 phosphorylation (Figure 6C,D). Deregulation of the AKT1 signaling pathway in the LL35 depleted cells, which participate in the regulation of glycolysis and gluconeogenesis, may explain the observed changes in the expression of the genes related to glucose metabolism.

### 3.6. Depletion of LL35 lncRNA Decreases Cell Survival and Proliferation In Vitro

Glucose and lipids’ metabolism plays a central role in cell biogenesis and is essential for cell life and proliferation. To reveal the role of LL35 lncRNA in cell survival and proliferation in vitro and analyze the data from transcriptome and proteome analysis regarding changes in the cell cycle pathways, we performed AN MTS assay for AML12 cells with depleted LL35 (Figure 7A). At day 2 after LL35 knockdown, we observed a decrease in cell survival in comparison with the control. At day 3 the difference between the cells with depleted LL35 and the control cells reached two times. To study the influence of LL35 depletion on the cell migration ability, we conducted a wound healing assay (Figure 7B,C). On the third day we saw that the wound closing rate was significantly lower in cells with LL35 knockdown compared with the control, indicating that LL35 downregulation impairs cell migration ability. E-cadherin protein participates in the formation of adherent junctions to bind cells with each other, which is directly related to the ability to migrate. We checked its expression on the mRNA level and found that LL35 depletion results in E-cadherin downregulation (Appendix A). Next, we performed a cell cycle analysis using flow cytometry and PI staining (Figure 7D). The obtained data revealed that LL35 knockdown leads to cell cycle arrest in the S phase (13% of cells with LL35 knockdown versus 9% of control cells), which may result in decreased cell proliferation. 

Finally, we analyzed the expression of multiple genes involved in important signaling pathways connected with cells’ proliferation and migration, Notch and NF-κB, after LL35 depletion. 

The Notch signaling pathway plays an important role in cellular proliferation, differentiation and apoptosis. The indirect immobilization of NOTCH ligand Jagged1 significantly reduced cell proliferation, colony forming unit ability and the number of cells in the S phase [49,50]. We demonstrated that LL35 knockdown results in the significant upregulation of Jagged1 protein, one of the main ligands, which binds to NOTCH receptors and triggers Notch signaling (Figure 8A). Moreover, NOTCH1 and NOTCH4 were also upregulated at mRNA level, as well as Notch signaling target genes—HES-1 and Hey-1 (Figure 8B). Thus, we can conclude, that LL35 contributes in normal Notch pathway signaling, which also affects the proliferation and migration of hepatocytes.

The NF-κB pathway is one of the main controllers of transcription, cytokine production and cell survival [51]. We measured the p105/p50 proteins’ ratio after LL35 knockdown and observed its slight increase (*p*-value = 0.06) (Figure 8C). However, there is an alternative way to activate the NF-κB pathway, via stimulation of the IκB (inhibitor of κB) kinases (Figure 8D,E). We evaluated the expression levels of IκBα mRNA and protein and observed their significant upregulation in two days after LL35 knockdown, which also contributes to the regulation of the cells’ proliferation and migration.

## 4. Discussion

Molecular mechanisms of action for most lncRNA remain unknown, but accumulating evidence suggests that the dysregulation of many lncRNA promotes various diseases [52]. For example, the expression levels of more than 100 lncRNA correlate with progression and survival for ovarian cancer, prostate cancer, hepatocellular carcinoma, fibrosis, glioblastomas and lung squamous cell carcinomas [53]. A lot of ncRNA functions are still undiscovered because of the high diversity of the possible mechanisms of action: the regulation of gene expression by interacting with proteins or DNA/RNA. Quite often the activity of lncRNA is associated with their unique tissue and subcellular localization and can also depend on the external cellular signals [54]. Expanding the opportunities for lncRNA investigation through the search of functional analogs in the other species and the use of animal disease models may help to uncover fundamental roles of lncRNA in vivo and their roles in disease development.

LncRNA LL35/Falcor is a proposed functional analog of human lncRNA DEANR1/linc00261 [22,23]. Linc00261 expression is downregulated in HCC, pancreatic, gastric, colorectal, lung and breast cancers, playing a role of tumor suppressor [17]. Based on published results, linc00261 RNA was proposed as a prognostic biomarker for HCC [55,56], for endometrial carcinoma [14] and gallbladder cancer [57]. We checked the LL35 RNA level in murine HCC tissue and found its downregulation, similar to human linc00261 (Figure 1B); moreover, LL35 levels in several murine hepatocyte cell lines additionally proved the predominant expression of LL35 in normal liver cells compared to hepatoma cells (Figure 1A). 

To investigate LL35 lncRNA functions in hepatocytes in vitro and in vivo we generated ASO-mediated knockdown of LL35. Conjugation of ASO with N-acetylgalactosamine (GalNAc) has become a breakthrough approach for liver-targeted delivery in the therapeutic oligonucleotide field [58] and allowed us to reach more than 80% downregulation of LL35 in the murine liver after a single administration of GalNAc-ASO without any damage confirmed by blood and morphological analysis. This result is consistent with previously shown data that the conditional knockout of LL35 lncRNA in mice results in relatively normal development, including endodermal-derived organs such as the lungs [23]. 

Transcriptome analysis of LL35-depleted hepatocytes revealed that LL35 downregulation causes changes in biological pathways related to metabolism, glutathione conjugation, biological oxidations and sulfur amino acid metabolism both in vitro and in vivo (Figure 2B,C). At the same time, we found changes in the cell cycle pathway only in the AML12 cell line that may be explained by the slow turnover of hepatocytes in the liver [43]. Despite some similar changed pathways, we found poor overlap between differentially expressed genes after LL35 depletion in cells and the murine liver, which can be explained by GalNAc-targeted delivery in hepatocytes only, while for transcriptome analysis, we used the whole part of the liver that contains other cell types. Previously, RNA-seq analysis was performed for lung epithelium isolated from null LL35 mice [23]. We did the same PANTHER Reactome pathway analysis of the published data for LL35 deletion in murine lungs and found similar changes in the cell cycle, cell adhesion and metabolism of amino acids pathways (Appendix A). Analysis of the published transcriptome data for human functional analog DEANR1/linc00261 depletion in ESCs demonstrated changes in the cell cycle, glutathione metabolism, cell adhesion and several signaling pathways [59], and in the case of DEANR1/linc00261 overexpression in human lung adenocarcinoma cells [60]. 

For complete characterization of the alterations after LL35 depletion and evaluation of the data concerning metabolism changes from the transcriptome, we performed a lipidome and metabolome analysis. We found that the abundance of plasmanyl-/plasmenylphosphatidylcholines (O-PC/P-PC) decreased in both cells and the murine liver, while phosphatidylethanolamines (PE) and hexosylceramides (HexCer) increased only in cell culture, and diradylglycerols (DG), ceramides (Cer;O2) and lysophosphatidylcholines (LPC) showed significant increase in mice (Figure 4B,C). Metabolome analysis revealed only changes in purine and glutathione metabolism and amino acids’ biosynthesis in vitro were affected by LL35 depletion (Figure 4E). In addition, we found changes in the mRNA expression of genes involved in lipid biosynthesis (Figure 4D). Growing studies showed that several lncRNA are involved in lipid metabolism via mediation of the expression of key genes and pathways that participate in the biosynthesis of cholesterol and triglyceride, and its transport and lipid uptake [61]. For example, Liu et al. showed that hepatocytes and diet-induced fatty liver levels of lncRNA H19 were upregulated by fatty acids, which led to an increase in triacylglycerol accumulation [62]. Meng et al. described that GAS5 lncRNA is highly expressed in THP-1 (macrophage-derived foam cells in coronary heart disease). The GAS5 depletion in THP-1 and homozygous ApoE (apolipoprotein E) knockout mice inhibited intracellular lipid accumulation and increased cholesterol efflux via reducing the EZH2-mediated transcriptional inhibition of ABCA1 through histone methylation [63]. Another lncRNA lncLSTR (liver-specific triglyceride regulator) functions in mammalian lipid homeostasis via the bile acid pathway. The depletion of lncLSTR in the mouse liver reduced plasma triglyceride levels [64]. Hence, the exact molecular mechanisms of lncRNA action and regulation of lipid biosynthesis and metabolism are still under investigation. 

Glucose metabolism is tightly connected with lipid metabolism. Excess glucose is stored in the liver as glycogen or is converted into fatty acids under insulin assistance and stored as a fat in adipose tissue. In the case of an overabundance of fatty acids, fat also builds up in the liver [65]. We found a significant reduction in the extracellular acidification rate (ECAR) level, which is an indicator of glycolysis, and the dysregulation of multiple genes involved in the main pathways of glucose metabolism, such as glycolysis, gluconeogenesis, pentose phosphate pathway, glutaminolysis, glycogen synthesis and acetyl-CoA synthesis in vitro (Figure 5B,C). We performed an insulin tolerance test in vivo to prove the relation between LL35 depletion and glycolysis and found that glucose blood levels in mice with LL35 KD were 1.5–2 times higher than in control mice (Figure 6A), which confirms a poorer insulin response. Analysis of the animal blood after the insulin injection demonstrated the compensation of the effects found for the LL35 depletion: a decrease in direct bilirubin and elevated levels of cholesterol, inorganic phosphorus and AST/ALT ratio (Figure 6B). For human functional analog DEANR1/linc00261, there are published data concerning the involvement of lncRNA in the regulation of glycolysis in pancreatic cancer. Zhai et al. demonstrated that linc00261 overexpression reduced the extracellular acidification rate (ECAR) and oxygen consumption rate (OCR). The authors proposed the role of linc00261 as a ceRNA to sponge miR-222-3p and activate the HIPK2-mediated ERK/c-myc pathway and also as a molecular decoy by sequestering IGF2BP1, which decreased the stability of c-myc mRNA and thus reduced c-myc expression. Such a reduction may influence the dysregulation of glycolysis [13]. However, the exact molecular mechanism of this regulation is still unclear. 

Our data demonstrated that LL35 depletion also causes a decrease in cells’ proliferation and cell cycle arrest in the S phase after KD of LL35 (Figure 7A,D), along with impairment of the migration ability (Figure 7B,C) in AML12 murine hepatocytes. To explain such phenotype changes we checked the status of the main signaling pathways such as the Notch and NF-κB pathways in LL35-depleted hepatocytes. We found that LL35 depletion results in significant upregulation of Jagged1 protein, one of the main ligands, which binds to NOTCH receptors and triggers Notch signaling, and the expression levels of IκBα mRNA and protein, which indicates the involvement of LL35 in the regulation of the NF-κB pathway (Figure 8). For the human functional analog, DEANR1/linc00261 participation in the regulation of the multiple signaling pathways such as Wnt/β-catenin signaling pathway, p38 MAPK signaling pathway, and Notch signaling pathway was demonstrated [21,55,66]. Studies of DEANR1/linc00261 were mainly conducted in the cancer cells, in which the overexpression of lncRNA inhibits cell proliferation and both cell invasion and migration by various ways [13,14,17,21,67,68]. Summing up, LL35/linc00261 is involved in the regulation of cell proliferation and migration by different mechanisms both in cancer and non-cancerous cells, while most molecular mechanisms are still under investigation. 

## 5. Conclusions

Taking our data together, we evaluated the predominant LL35 RNA expression in normal hepatocytes and the involvement of murine lncRNA LL35 in the regulation of the glycolysis and lipid biosynthesis in vitro and in vivo. We demonstrated downstream effects from LL35 depletion in normal hepatocytes on the Notch and NF-κB pathways, and all these changes of the transcriptome, lipidome and signaling pathways lead to the strong phenotype in hepatocytes—a decrease in proliferation, migration and cell cycle arrest. All obtained data correlate with the published data for LL35/Falcor transcriptome in the lungs and for human functional analog DEANR1/linc00261. Although linc00261 was first identified several years ago [69], its roles as a novel lncRNA involved in a variety of diseases and its molecular mechanisms are still a target of investigations. Results and functions for the murine LL35 lncRNA in the animal models of human diseases can be translated to human functional analog linc00261 and expand the potential therapeutic role of human lncRNA. 

Studies of species-specific lncRNA in vivo, which are functional analogs of human ones, may broadly highlight the involvement of lncRNAs in certain biological processes and diseases. 

We propose that studying LL35 in an HCC mouse model will be helpful in uncovering the exact molecular mechanisms in vivo.

## Figures and Tables

**Figure 1 biomedicines-10-01397-f001:**
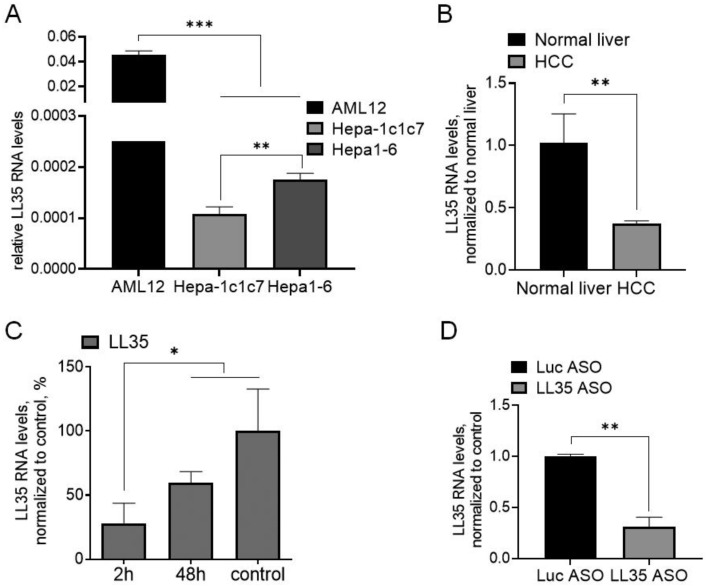
Evaluation of LL35 RNA levels in vitro and in vivo by RT-qPCR. (**A**) Hepatocyte cell lines. (**B**) Normal and HCC mouse liver. (**C**) Partial hepatectomy. (**D**) LL35 KD in the liver at day 5 after injection of GalNac-ASO conjugates (LL35 ASO). Luc ASO—control. ACTB—reference gene in RT-qPCR. Results show mean ± SD. * *p* < 0.05, ** *p* < 0.01 and *** *p* < 0.001.

**Figure 2 biomedicines-10-01397-f002:**
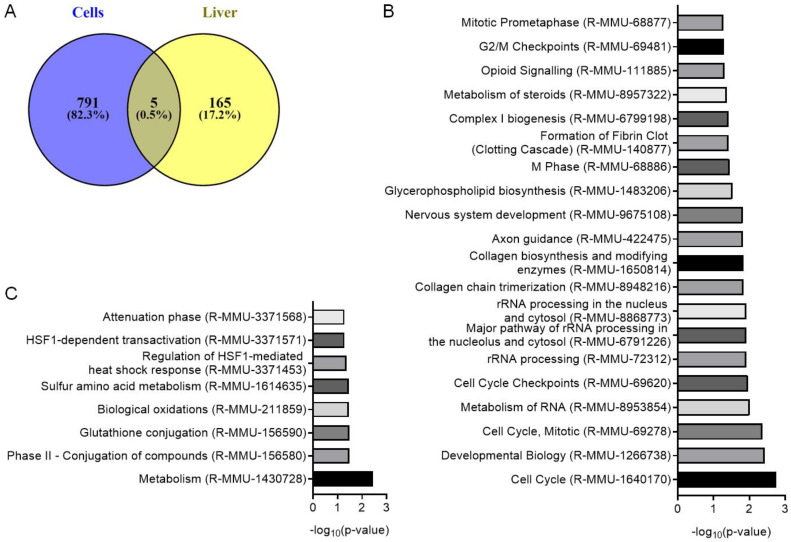
Results of transcriptome analysis of AML12 cells and murine liver after LL35 depletion. (**A**) Venn diagram for differentially expressed genes after LL35 knockdown in AML12 cells in comparison with murine liver. Top representative pathways with *p*-value < 0.05 obtained by PANTHER Reactome analysis of differentially expressed genes after LL35 depletion in (**B**) AML12 cells; (**C**) murine liver.

**Figure 3 biomedicines-10-01397-f003:**
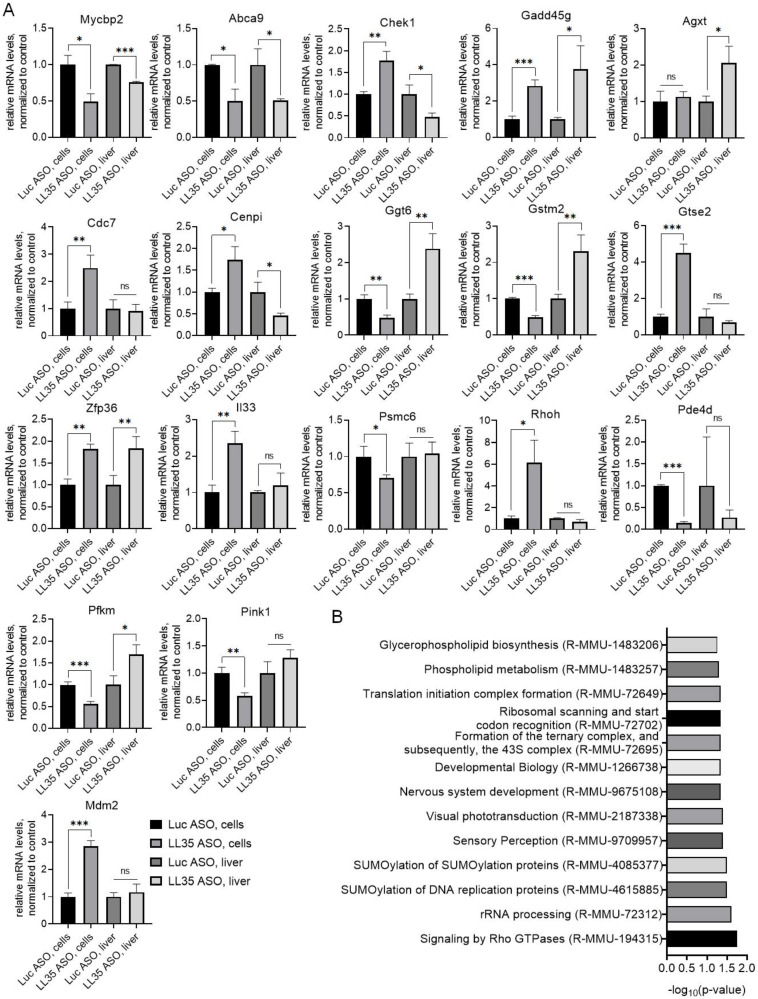
Transcriptome validation and proteome analysis of AML12 cells and murine liver after LL35 depletion. (**A**) Validation of transcriptome data by RT-qPCR, normalization on ACTB gene. (**B**) Top representative pathways obtained by PANTHER Reactome analysis for significantly changed proteins after LL35 depletion in AML12 cells. Results show mean ± SD. ns—not significant. * *p* < 0.05, ** *p* < 0.01 and *** *p* < 0.001.

**Figure 4 biomedicines-10-01397-f004:**
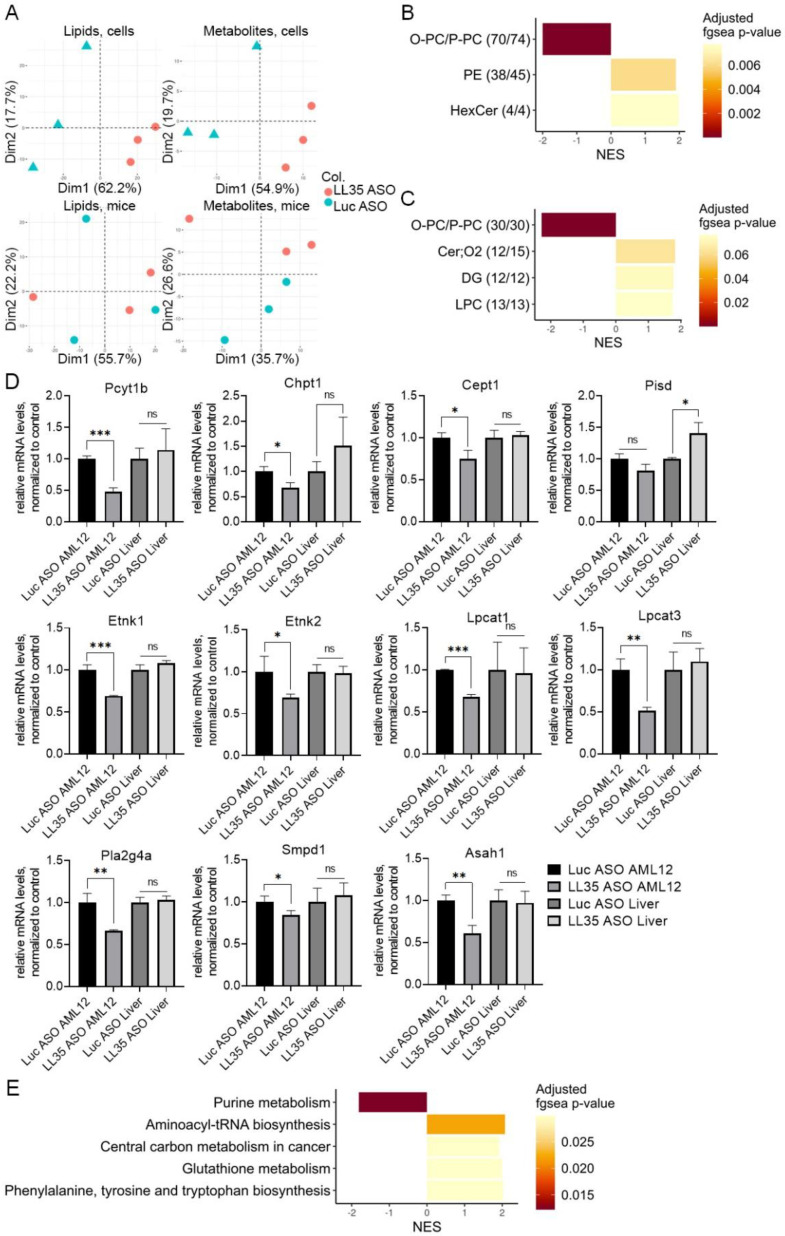
Lipidome and metabolome analysis of AML12 cells and murine liver after LL35 inhibition. (**A**) PCA of lipids and metabolites from AML12 cells and liver samples with depleted LL35 (LL35 ASO) and control (Luc ASO), polarities merged. Lipid abundance alteration in (**B**) AML12 cells; (**C**) murine liver, classes that are significant both before and after confirmation (fgsea FDR < 10%). Numbers in brackets: the number of confirmed features and the total number of features. (**D**) RT-qPCR analysis of expression levels of key genes, which participate in synthesis and breakdown of significantly changed lipids classes after LL35 knockdown (LL35 ASO) and control (Luc ASO), normalization on ACTB gene. (**E**) KEGG pathways, metabolites from which are significantly influenced by LL35 knockdown in AML12 cells. NES—normalized enrichment score. Results show mean ± SD. ns—not significant. * *p* < 0.05, ** *p* < 0.01 and *** *p* < 0.001.

**Figure 5 biomedicines-10-01397-f005:**
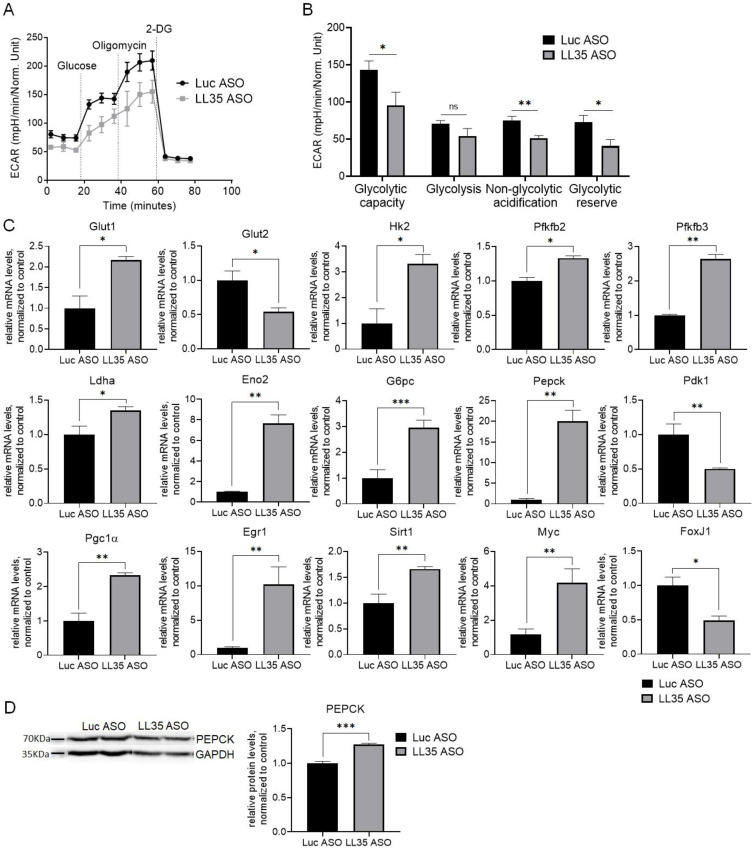
Changes in the glucose metabolism after LL35 knockdown in vitro. (**A**) An averaged time curve for ECAR after subsequent injections of 10 mM glucose, 1 μM oligomycin and 50 mM 2-DG. Each data point represents an ECAR value used for calculations. (**B**) Individual parameters for cell glycolytic function, including glycolytic capacity, glycolysis, non-glycolytic acidification and glycolytic reserve, for AML12 cells after LL35 depletion (LL35 ASO) and control (Luc ASO). (**C**) Evaluation of mRNA levels for key genes and transcription factors, which participate in glucose metabolism after LL35 knockdown (LL35 ASO) and control (Luc ASO) by RT-qPCR with normalization on ACTB gene. (**D**) Western blot and its quantification of PEPCK protein levels in AML12 cells after LL35 inhibition (LL35 ASO) and control (Luc ASO). Results show mean ± SD. ns—not significant. * *p* < 0.05, ** *p* < 0.01 and *** *p* < 0.001.

**Figure 6 biomedicines-10-01397-f006:**
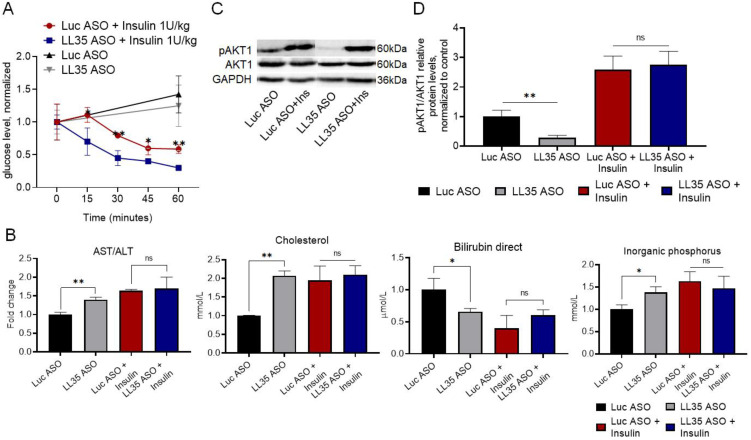
The influence of insulin in mediating the effects of LL35 depletion. (**A**) Insulin tolerance test in vivo. Glucose blood level in mice was measured each 15 min for 1 h after insulin injection (1 U/kg). LL35 ASO—mice with depleted LL35, Luc ASO—control mice. (**B**) Biochemical parameters of mice blood at day 5 after LL35 depletion (LL35 ASO) in comparison to control mice (Luc ASO). Insulin treatment compensates these differences. (**C**) Western blot of pAKT1 and AKT1 proteins in AML12 cells after LL35 knockdown (LL35 ASO) and control (Luc ASO) with 100 nM (+Ins) or without insulin treatment. (**D**) Quantification of pAKT1/AKT1 ratio for AML12 cells after LL35 knockdown and insulin treatment. Results show mean ± SD. ns—not significant. * *p* < 0.05, ** *p* < 0.01.

**Figure 7 biomedicines-10-01397-f007:**
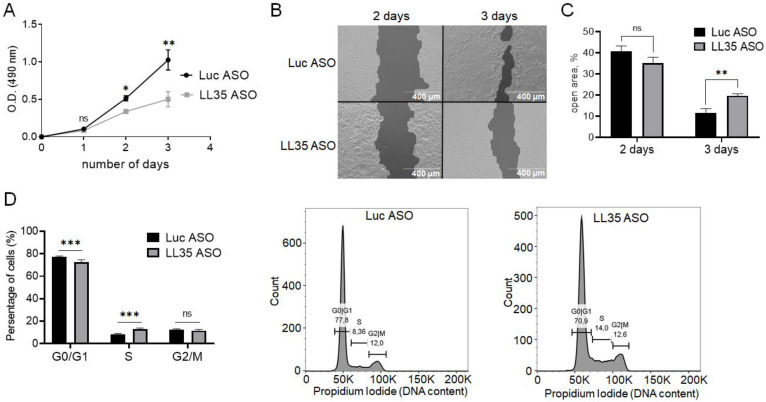
(**A**) Study of hepatocyte survival after LL35 depletion (LL35 ASO) in comparison with control cells (Luc ASO). (**B**) Wound healing assay for AML12 cells after LL35 knockdown and control. (**C**) Quantification of wound healing assay with ImageJ software. (**D**) Cell cycle analysis of the cells with LL35 knockdown and control cells by flow cytometry. Percentage of cells in different cell cycle phases after LL35 knockdown compared with control obtained from flow cytometry analysis after PI staining. Results show mean ± SD. ns—not significant. * *p* < 0.05, ** *p* < 0.01 and *** *p* < 0.001.

**Figure 8 biomedicines-10-01397-f008:**
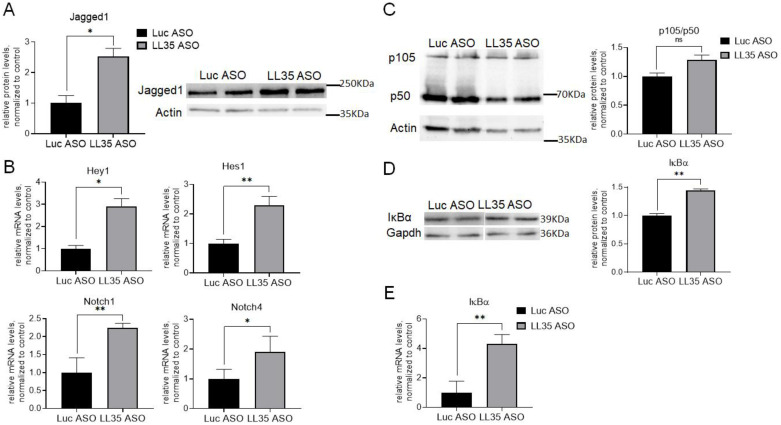
Changes in Notch and NF-κB pathways after LL35 depletion in vitro. (**A**) Estimation of Jagged1 protein level after LL35 knockdown in vitro by western blot, normalized at ActB protein level. (**B**) RT-qPCR measurement of the mRNA expression levels of major genes involved in Notch signaling pathway after LL35 knockdown (LL35 ASO) in AML12 cells and control (Luc ASO), normalization on ACTB gene. (**C**) Estimation of p105/p50 proteins’ ratio after LL35 knockdown in AML12 cells by western blot, normalized on ActB protein level. (**D**) Estimation of IκBa protein level after LL35 knockdown by western blot, normalization on GAPDH protein level. (**E**) RT-qPCR measured expression levels of IκBa mRNA after LL35 knockdown and control. ACTB—reference gene. Results show mean ± SD. ns—not significant. * *p* < 0.05, ** *p* < 0.01.

## Data Availability

The tables for gene expression, metabolite and lipid abundances are included in the Appendix A. The mass spectrometry proteomics data have been deposited to the ProteomeXchange Consortium via the PRIDE [70] partner repository with the dataset identifier PXD033910. The transcriptomic data discussed in this publication have been deposited in NCBI’s Gene Expression Omnibus [71] and are accessible through GEO Series accession number GSE203431 (https://www.ncbi.nlm.nih.gov/geo/query/acc.cgi?acc=GSE203431).

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
