# Peer review of "Murine Falcor/LL35 lncRNA Contributes to Glucose and Lipid Metabolism In Vitro and In Vivo"

_biomedicines, 2022, doi:10.3390/biomedicines10061397_

Round 1

Reviewer 1 Report

In the present study, the Authors investigated the biological role of lncRNA Falcor/LL35 in murine hepatocytes in physiological as well as pathological conditions, demonstrating (in vitro and in vivo) the involvement of this non-coding RNA in the regulation of crucial metabolic processes, such as glycolysis and lipid biosynthesis, investigated the changes in the transcriptome, lipidome and metabolome upon LL35 depletion and provided an interesting parallelism with similar results observed for another lncRNA, human linc00261, thus opening the possibility of translating their results in the human pathology.

Overall, the Article is well-written, the topic is interesting and well explained. The Introduction is clear, ending with an explanation of the main objectives and results obtained. The Methods section is very detailed, with a clear explanation of the methodologies applied and statistical analyses used. Results are organized in a logic order and the conclusion deduced by the data are interesting. The final Discussion section is well carried out summarizing the results obtained in a proper methodological and biological context by a sufficient comparison with the current background and highlighting additional investigation possibilities.

Overall, this is an excellent work. I have very few suggestions:

1)      I noticed few typos that can be easily corrected with a final reading;

2)      Western blotting figures: scale bars are missing and should be reported, as done in the supplementary;

3)      It is not clear how many animals have been used for the experiments reported; please, specify this in M&Methods;

4)      Figure 2B and 2C; for the gene ontology, the significance threshold (p value) must be reported (for example, as done for the volcano plot is Supplementary Fig. S3);

5)      Since the Article is very rich of data, a graphical abstract showing the multiplicity of cellular effects mediated by LL35 and its connection with crucial cellular pathways would be useful to the reader;

6)      Is it LL35 also involved in other cancers, besides the HCC? Do the Authors evaluated if it analogue, human linc00261, is differentially regulated in other human cancers? This point could be of interest for the scientific community, giving a higher impact to the work, and could be easily explored not only in literature but also in TCGA data; indeed, a good candidate biomarker is also characterized by specificity.

Author Response

We would like to thank you for your careful review of our manuscript and for the time you’ve spent reviewing it. Your comments and remarks were very valuable and served to improve the quality of our manuscript. We have made changes to the manuscript according to your comments in edit mode (highlighted in green) and also below you can find our answers to your comments.

1) During the editing of the manuscript, we found several typos and corrected them.

2) Thank you for this point, we added scale bars to the all figures with western blotting membranes in the main text (Figures 5D, 6C, 8A, 8C, 8D)

3) For each in vivo experiment we used 3 animas per condition. We added a few more clarifications about the number of animals in the Materials and Methods section of the manuscript.

4) After we performed a pathway enrichment analysis for differentially expressed genes, we truncated the resulting pathway lists by p-value<0.05. Only pathways with p-value<0.05 are shown on the Figures 2B and 2C for in vitro and in vivo experiments. We have added a description of threshold p-value for these two graphs in the text of the manuscript.

4) Thank you for this suggestion, we drew a graphical abstract and submitted it together with the manuscript.

5) In our previous paper (Sergeeva et al., 2019) we estimated LL35 expression level in different murine tissues and found its predominant expression in the liver and lungs. Also LL35/Falcor function was previously studied in the normal lung tissue because of the high RNA expression. Meanwhile, multiple publications demonstrated the involvement of human linc00261 in the cancer progression. For example, linc00261 is downregulated in HCC, choriocarcinoma, pancreatic, gastric, colorectal, lung and breast cancers, compared to corresponding normal tissue (Zhang et al., 2021) and proposed as a potential biomarker for endometrial carcinoma (Fang et al., 2018), gallbladder cancer (Niu et al., 2020). Published data confirm the relevance of the search and study of the murine functional analogue for the deep understanding linc00261 functions. Also we added several sentences about linc00261 expression in other cancer types in the Introduction part of the manuscript.

Reviewer 2 Report

The Manuscript: „ Murine Falcor/LL35 lncRNA contributes in glucose and lipid metabolism in vitro and in vivo’’ by Evgeniya Shcherbinina and colleagues demonstrate the prevalent expression of lncRNA LL35 in normal hepatocytes in comparison to cancer and proliferating cells in vitro and in vivo, based on the experiments performed on different cell lines and animal model. Over the years, long noncoding RNAs (lncRNA) have been known to play important roles in the regulation of transcription, splicing, translation, and other processes in the cell and various studies have demonstrated that the expression of transcription Factor Foxa2 is extremely regulated by lncRNA in Normal and Fibrotic Mouse Liver. The present study further emphasizes the importance of lncRNA in metabolism, especially in metabolism of glucose and lipid metabolism in vitro as well as in vivo. The study is nicely conducted with elaborate description of methodology and documentation of subsequent result. After going through the manuscript, I have following comments to the authors:

1.   The methodological part is missing in abstract. Please include a one-line description of the methodology.

2.       The material and method section is too bulky with the detailed description of each methods. I would suggest including only the important information in the manuscript and all other details can be documented in a separate supplementary file.

3.       The discussion section is also bulky with repetition of results at many instances (for example: lines 803-804: Next, we generated ASO-mediated knockdown of LL35 lncRNA in hepatocytes in vitro and in vivo to investigate its functions; lines 811-812: Then we compared changes in gene expression by transcriptome and proteome analysis in vitro and in vivo; and so on.) Please trim the discussion to make it clear and to the point.

4.       Please discuss briefly the therapeutic importance of the study and how the outcomes of the study could be implemented in improving the quality of life in humans.

Author Response

The authors and I would like to thank you for your careful review of our manuscript and for the time you’ve spent reviewing it. We highly appreciate your comments and remarks, which have greatly served to improve our manuscript. Below you will find our answers to your comments, and all required changes was made directly in the text of the manuscript in edit mode (highlighted in green).

1) Thank you for this point, we included a short description of the methodology in the abstract of our manuscript.

2) We agree with your point of view and have shortened the Materials and Methods section by moving part of the description to Supplementary Materials. In particular, we trimmed the following sections: western blotting, RNA-seq data processing and analysis, LC-MS-based proteomics, cell lines and liver extraction protocol for lipids and metabolites analysis, liquid chromatography/mass spectrometry analysis of lipidome, liquid chromatography/mass spectrometry analysis of metabolome, post-acquisition processing and statistical analysis of lipidome and metabolome data.  

3) Thank you for this suggestion, we have trimmed the Discussion section of the manuscript.

4) LL35 human analog lncRNA linc00261 is differentially expressed in different cancer types, such as HCC, choriocarcinoma, pancreatic, gastric, colorectal, lung and breast cancers (Zhang et al., 2021). Linc00261 serves as a tumor suppressor and in some cases as a biomarker for patient response to chemotherapy (Lin et al., 2019). In our study we showed that obtained results for murine LL35 correlate with previously published data for linc00261 in hepatocytes. This correlation gives an opportunity to translate the results of LL35 in vivo studies to human linc00261. We added a discussion about the therapeutic importance of our study in the Discussion section of our manuscript.